# Cardiac Evaluation of Exercise Testing in a Contemporary Population of Preschool Children: A New Approach Providing Reference Values

**DOI:** 10.3390/children9050654

**Published:** 2022-05-03

**Authors:** Pedro Ángel Latorre-Román, Martínez Martínez-Redondo, Jesus Salas-Sánchez, Pedro José Consuegra-González, Elena Sarabia-Cachadiña, Jerónimo Aragón-Vela, Juan A. Párraga-Montilla

**Affiliations:** 1Department of Corporal Expression, University of Jaen, 23071 Jaen, Spain; platorre@ujaen.es (P.Á.L.-R.); consuegragonzalezpj@gmail.com (P.J.C.-G.); jparraga@ujaen.es (J.A.P.-M.); 2Consejería de Educación de Andalucía, CEIP Doctor Fleming, 23500 Jódar, Spain; melchor_mr@hotmail.com; 3Universidad Autónoma de Chile, Santiago 7500912, Chile; jsalas@ujaen.es; 4Department of Physical Activity and Sport, Cardenal Spínola-CEU University Studies Center, 41930 Sevilla, Spain; esarabia@ceuandalucia.es; 5Department Physiology, Faculty of Sport Sciences, University of Granada, 18011 Granada, Spain

**Keywords:** health care, physical activity, sport physiology, gender, exercise

## Abstract

The objective of this study was to evaluate cardiac autonomic function at rest, during maximal exercise, and in post-exercise recovery, to determine sex-specific and age-specific differences in resting heart rate (RHR), linear and spectral parameters of Heart Rate Variability (HRV), HR_peak_, and heart rate recovery (HRR) after one and five minutes, in preschool children. This study involved a cohort of 167 healthy children (79 girls) aged 3 to 6 years that were selected from several schools in southern Spain. A 10 × 20 m test was conducted, and the cardiovascular response was recorded. No significant differences were found in all variables between the sexes. However, a significant reduction in RHR and an increase in HRR were found from age 4 to age 6. HRV parameters at rest were higher in older children. No associations between 10 × 20 m performance, weight status, and cardiac parameters were found. Simple linear regression analysis revealed that heart rate reserve (HRr), HRR_5min_, RMSSD, and HF were the variables that showed association with all HR parameters. There was also a significant correlation between HRr and HRR_5min_. In conclusion, cardiovascular autonomic function during rest, exercise, and recovery in Spanish preschool children was not influenced by sex, although older children showed greater cardiovascular modulation. Cardiorespiratory fitness status was not associated with HR response.

## 1. Introduction

Childhood obesity is an alarming problem in industrialized countries. Even young children, such as preschoolers, present a remarkable rate of obesity. Physical activity (PA) has an important role in the quality of life and health of these children since it has been demonstrated that the incidence of obesity is lower in active kids [1]. Gaining weight is not the only problem for nonactive infants; the lifestyle which provokes the extra weight also brings other health issues associated with cardiometabolic risk (CMR), a lower function of the cardiac autonomic nervous system, and lower cardiorespiratory fitness (CRF) [2]. However, when children participate in PA programs, positive health effects have been noted, such as high cardiovascular function, low adiposity, high physical fitness, and well-being [3]. Bürgi et al. indicated that in preschool children, PA level is associated with improvements in CRF [4]; CRF is a relevant biomarker of health [5].

Cardiovascular complications have raised public health costs around the world, and it has been demonstrated that most of these complications, which appear in adulthood, originate in childhood [6]. Throughout the maturation process (i.e., from babyhood to adulthood), some variations in the modulation of the autonomic nervous system over the heart occur, inducing changes in heartbeat patterns. In addition, the maturation process leads to important morphological and functional changes that can significantly modify the responses of the heart to exercise [7]. Hence, abnormal exercise performance in children might be an early indicator of cardiovascular dysfunction, which can be examined through a validated exercise test [8]. 

Heart rate (HR) can be used to detect differences in cardiovascular health in preschool children by measuring the cardiovascular response to physical exercise with an uncomplicated clinical assessment [6]. Therefore, HR is a biomarker that can be utilized to determine physiological function and health status. In this regards, the analysis of the HR reaction to exercise can clarify the physiological mechanisms involved in physical exercise, but some of these parameters are not applicable to the pediatric population due to a lack of research in this area [9].

Specifically, children who have a resting heart rate (RHR) similar to or superior to 91 beats per minute (bpm) show higher mean LDL cholesterol [10] and an increase in blood pressure, which are independent of adiposity, ethnicity, and age [11]. Otherwise, bradycardia can be explained as an RHR that is slower than the normal RHR for a specific population’s age [12], which is considered an RHR of <60 bpm [13], and is related in pediatric patients with increased vagal tone, hypothyroidism, hypothermia, adrenal insufficiency, increased intracranial pressure, inherited arrhythmia, and cardiomyopathy or complete heart block and can lead to sudden death [14].

On the other hand, during exercise, a lower HR peak is usually defined as chronotropic incompetence when <80% of the predicted value is reached during exercise [15]. The clinical relevance of chronotropic incompetence in children is still unclear, as this is present many years before heart disease develops [16]. Furthermore, the chronotropic incompetence can be an index of exercise intolerance [17].

Once PA has been finished, cardiac output is reduced by parasympathetic reactivation and sympathetic inhibition since HR returns to resting levels gradually. These modifications can be assessed by HR recovery (HRR). In children, a low HRR after maximal exercise also suggests impaired parasympathetic function, which is found in children with sickle cell anemia [18], overweight [19], and obese children with metabolic syndrome [20].

Moreover, it is also interesting to analyze heart rate variability (HRV) in children as a method for determining autonomic nervous system function. HRV is defined as the variability of the distance between consecutive R peaks of the ECG signal [21]. Low HRV in children may be related to some cardiac and non-cardiac dysfunctions in a variety of clinical conditions such as congenital heart disease [22], pathogenesis of allergic diseases [23], autism spectrum disorder [24], diabetes mellitus [25], and attention-deficit/hyperactivity disorder [26].

Therefore, the assessment of the response to exercise is a significant clinical tool since it supplies an evaluation of the cardiorespiratory function [27], which is relevant in preschool children. In this regard, CRF assessment provides relevant information in terms of the lifestyle and health of the children, detecting pathological alterations that do not appear in other resting clinical exams [28]. Nevertheless, despite this topic being very relevant, the research on cardiovascular autonomic function in preschool children is sparse [29]. To adequately understand cardiovascular autonomic function in preschool children under these circumstances, it is essential to determine if HR changes depend on age and sex, which could be related to the maturation of the cardiovascular autonomic function. Thus, the main aim of the present work was to analyze autonomic nervous system function under three different conditions: rest, maximal exercise, and recovery using the 10 × 20 m test in healthy Spanish preschool children. In addition, this study also aimed to investigate the association between HR, cardiorespiratory fitness, and anthropometric markers in this population.

## 2. Materials and Methods

### 2.1. Participants

The G*Power software was used to calculate the number of participants in this research [30]. The resulting variables were moderate effect size (*f*  = 0.325), α level of 0.05, a power level of 0.95, 4 groups, 1 measurement, and critical F = 2.660. Finally, the number of participants was determined to be at least 167 children. Consequently, this cross-sectional study included 167 healthy children (79 girls and 88 boys) from 3 to 6 years (age = 4.68 ± 1.21 years). The study selected the population from different rural and nonrural schools in Andalusia (southern Spain). Children with chronic diseases, including metabolic or endocrine disorders, diabetes, asthma, and any condition that interfered with physical exercise, were excluded from the study. In addition, to participate in the study, a voluntary consent form was signed by the parents of each participant involved in the research. Moreover, our study followed the ethical recommendation approved in the Declaration of Helsinki (2013). Also, the investigation was approved by the Ethics Committee of the University of Jaén (Spain).

### 2.2. Materials and Testing

#### 2.2.1. Anthropometric Variables

Body mass (kg) was measured using a weighing scale (Seca 899, Hamburg, Germany) and body height (cm) with a stadiometer (Seca 222, Hamburg, Germany). Body mass index (BMI) was calculated by dividing body mass (kg) by body height^2^ (in meters). Waist circumference (WC) was measured at the umbilical location using a non-elastic Ergonomic Circumference Measuring Tape (Seca 201, Hamburg, Germany).

#### 2.2.2. Cardiorespiratory Analysis

To provide a comprehensive analysis of cardiovascular health, the study assessed 5 related indicators: CRF, RHR, HR_peak_, HRR, and HRV. CRF was measured using the 10 × 20 m test [31]. It was a 20 m shuttle test in which participants moved 5 balloons from a box, A, located at one extreme, to the other box, B, located at the opposite extreme. The total distance included was 200 m. The result was registered in seconds, and only one decimal was used. All the participants were encouraged to complete the test as quickly as possible.

For collecting HR data, the study used the R-R interval monitor Firstbeat Bodyguard 2 (Firstbeat Technologies Ltd., Jyväskylä, Finland), which works as a one-channel ECG recorder at 1000 Hz of sample rate. It takes the signal through two electrodes located in the chest, and its feasibility has been proven in the literature [30,31]. The Software Firstbeat Sports (Firstbeat Technologies Ltd., Jyväskylä, Finland) performed HRV analysis for frequency and time-domain parameters. On one hand, the frequency domain analysis provided high-frequency (HF) and low-frequency (LF) spectrum, the HF being an indicator of parasympathetic activity while the LF can combine both sympathetic and parasympathetic modulation [32]. The study also calculated the ratio between the spectrums (LF/HF ratio), which is described in the literature as an indicator of sympathovagal balance [32]. Following Bobkowski et al. [33], this work avoided HF and LF in normalized units. On the other hand, this investigation selected two-time domain measures: the root-mean-square differences of successive heartbeat intervals (RMSSD) and the mean of the standard deviation of the NN interval (SDNN), which are indicators of parasympathetic modulation [34,35]. The data collection for HRV analysis followed the European Society of Cardiology and the North American Society of Pacing and Electrophysiology recommendations [35]. The previously mentioned software analyzed HR data as well, making possible the calculation of the rest of the variables. Thereby, the HR reserve (HRr) was calculated as follows: HRr = HR_peak_ at exercise-HR at rest, while the chronotropic index (CI) was CI = (HR_peak_ at exercise − resting HR)/[(220 − age) − resting HR)] [36]. Chronotropic incompetence was set when the CI was <0.80 [36]. Lastly, the calculation for HRR was set at minutes 1 and 5 of the recovery. Thus, the calculations were HRR_1m_ = HR_peak_ at exercise-HR at minute 1 of the recovery time; and HRR_5m_ = HR_peak_ at exercise-HR at minute 5 of the recovery time. As expressed in the literature, HRR_1min_ and HRR represent the fast and slow phases of recovery, respectively [37]. The study considered, in the same line as Paridon et al. [38], that a child performed a maximal exercise test when the HR_peak_ at exercise reached 200 beats/min.

### 2.3. Procedure

Before the exercise test, the study collected anthropometric parameters and the data for calculating all HR-related variables at rest. These data were obtained for 10 min, seated and breathing spontaneously in accordance with Latorre et al. [39]. To stabilize the HR, participants were told to stop talking or making any movement during the data collection. After that, the children performed the 10 × 20 m test without running, just walking as a trial of familiarization. Subsequently, the children completed different mobility exercises to increase basal status and performed the 10 × 20 m test. There was HR monitorization during the whole test to obtain HR_peak_, and in accordance with [40], all subjects went to sit for 10 min right after concluding the test to calculate HRR_1m_ and HRR_5min_. During this period, all participants were encouraged to sit and not to move, to breathe normally, and not talk with their partners.

### 2.4. Statistical Analyses

The software SPSS, v.22.0 for Windows (SPSS Inc, Chicago, IL, USA) performed the statistical analysis. The significance level was set at *p* < 0.05. Descriptive statistics are presented in terms of means and standard deviations (SD). Tests of normal distribution and homogeneity (Kolmogorov–Smirnov and Levene’s, respectively) were carried out. The Mann–Whitney U-test analyzed differences between boys and girls in HR parameters, whereas the Kruskal–Wallis test examined the differences among values from the 3 to 6-years age groups. A post-hoc analysis by the Mann–Whitney U-test determined differences between groups adjusted by the Bonferroni test (α = 0.05/4 = 0.01). Differences in RHR, HRr, CI, and 10 × 20 m performance were evaluated by analysis of variance (ANOVA) corrected by the Bonferroni test. Additionally, for verifying the correlation of HR variables with CRF and anthropometric variables, a partial correlation evaluation and a linear regression analysis of the associations between variables were performed (adjusted by age and sex). The magnitude of correlation among measurement variables was set according to Hopkins et al. [41].

## 3. Results

Table 1 shows anthropometric features, 10 × 20 m performance, and HR responses concerning sex and age groups. Regarding sex, boys had lower results of WC (*p* = 0.016) than girls. No significant differences (*p* ≥ 0.05) were found in the rest of the variables between the sexes. Regarding age groups, in all samples, significant differences were found (*p* < 0.05) in each variable analyzed, with an exception for BMI, HR_peak_, HRr, CI, and LF/HF. Furthermore, in both boys and girls, 10 × 20 m performance improved with age (*p* < 0.001) (Table 1).

A significant decrement in RHR was shown in both boys and girls between age groups from 3 to 6 years. However, the HR_peak_ did not show significant changes. Regarding HRR_1min_, significant increases were observed in both boys and girls between age groups from 3 to 6 years, though, in the HRR_5min_ significant increases were only observed between age groups from 3 to 6 years in girls but not in boys (Figure 1).

Regarding the interaction between age groups and sex, the study did not find any differences in HR variable (Figure 1), however, girls displayed higher values than boys in 10 × 20 m at age 4 (Table 1). An analysis of partial correlation indicated that RHR displayed significant correlation with HR_peak_ (*r* = 0.163; *p =* 0.036), HRr (*r* = −0.579; *p* < 0.001), HRR_1min_ (*r* = −0.246; *p* = 0.001), HRR_5min_ (*r* = −0.358; *p* < 0.001), and all HRV parameters (*p* < 0.001). In addition, HR_peak_ showed significant correlation with HRR_1min_ (*r* = 0.229; *p* = 0.003) and HRR_5min_ (*r* = 0.607; *p* < 0.001). HRR_1min_ displayed significant correlation with HRr (*r* = 0.365; *p* < 0.001) and CI (*r* = 0.274; *p* < 0.001). Moreover, HRR _5min_ showed significant correlation with HRR_1min_ (*r* = 0.757; *p* < 0.001), HRr (*r* = 0.757; *p* < 0.001), and CI (*r* = 0.678; *p* < 0.001). Finally, HRr displayed significant correlations with RMSSD (r = 0.483, *p* < 0.001), SDNN (r = 0.547, *p* < 0.001), HF (r = 0.386, *p* < 0.001), and LF (r = 0.315, *p* < 0.001).

Simple linear regression analysis showed that HRr, HRR_5min_, RMSSD, and HF were the variables that were associated with all HR parameters (Table 2). The Scatter plot for HRR_5min_ versus HRr is shown in Figure 2. A significantly positive association appeared between the two variables that were alike in both sexes. The R^2^ for the model was 0.654 and 0.492 for boys and girls, respectively.

Figure 2 shows a scatter plot between HRr and HRR_5min_. As can be seen there is a strong positive correlation between HRr and HRR_5min_ in both boys and girls.

## 4. Discussion

To our knowledge, this is the first study that provides reference values of autonomic cardiac function at rest, during maximal exercise testing, and during recovery for Spanish preschool children. The major findings of this study were: (1) RHR decreased with age and HRR, and HRV changed as age increased. (2) No significant differences among sexes were found in any cardiac variable. (3) No associations between 10 × 20 m performance, weight status, and cardiac parameters were found. 

The assessment of the physiological response to exercise is a significant clinical tool since it provides an evaluation of the cardiorespiratory system [27], and it has particular relevance in children aged 3–6. Greater levels of CRF in childhood and young adulthood can be related to a better health and cardiovascular profile in the future. [42]. In this regards, the 10 × 20 m test is a valid measurement of cardiorespiratory response to exercise in this age group [43]. Moreover, one of the advantages of the short-term recordings is that it allows obtaining reliable results in a shorter time, which is very useful and brings better feasibility for clinicians and PA teachers [21].

### 4.1. Cardiac Autonomic Function at Rest

RHR kept between 71–133 bpm range, these values being similar to those revealed previously [44,45], which ranged from 81–125 bpm. In the current study, age, HR_peak,_ HRr, HRR, and HRV influence RHR. Moreover, in the same line as the present results, a previous study demonstrated that RHR is negatively correlated with HRR [46].

Mainly, high RHR is associated with younger age. Our results are in accordance with those found in previous studies [45,47], which note a reduction of RHR in older children. Regarding sex, Rabbia et al. [48] suggested that RHR is higher in girls independently related to somatic growth indexes, PA, and socio-cultural level. Moreover, Ostchega et al. [45] revealed that mean RHR is significantly higher in girls than boys during childhood and adolescence, from age 1 to 19. However, in the current study, in accordance with Mimura and Maeda [44], no significant differences were found between sex and RHR.

On the other hand, RHR could be utilized to identify kids with a greater risk of developing cardiovascular threat factors [49]. Also, RHR ≥ 86 bpm is associated with an increased likelihood of high blood pressure in both non-obese and obese children [11]. However, in the present study, no relationship was found between RHR and WC or BMI. Our findings align with those of Kwok et al. [50], who noted a weak relationship between obesity and RHR.

Though it can be difficult to delimit an ideal RHR for a certain individual, according to Oschega et al. [45] and Hart [51], it seems desirable to maintain RHR among a recommended range of 86–109 bpm from ages 3 to 6. In the current study, although RHR was not recorded at basal condition, 67.6% of girls and 64.4% of boys achieved the target. 

### 4.2. Cardiac Autonomic Function during Maximal Exercise

Regarding HR_peak_, in the present study, various factors such as RHR, HRr, CI, HRR, and HRV influenced this parameter. In contrast to adults and children or adolescent populations [52,53], data regarding the HRmax prediction in preschool children are unknown. Given that there is a small HR_peak_ range in youth, Gelbart et al. [52] proposed 197 bpm as the mean HR_peak_ in infants and adolescents, with 180 bpm for the lowest value. Specifically, Van Brussel et al. [7] noted that the average HR_peak_ remains relatively stable, around 195 to 197 (bicycle) to 200 beats/min (treadmill), in children and adolescents. The average HR_peak_ achieved in the current study was near to this range and also agrees with a previous study which showed that preschool children displayed HR_peak_ values from 196–200 bpm [54]. In the present investigation, average values of CI in boys and girls achieved the target >0.80. It might be an indicator of a healthy HR response to exercise. However, von Sheidt et al. [36] showed a CI < 0.80 in healthy infants if compared to others with heart diseases due to congenital factors. Hence, the use of the 0.8 threshold is not reliable for detecting CI in treadmill exercise testing in infants, most likely due to an overestimation of the maximal HR using the (220-age) formula.

It is noteworthy to state that the findings of this investigation indicate that HRR_5min_ shows a significant large correlation between HRr and CI. About this issue, one study [55] indicates that these two parameters, HRR and CI, are related and are commonly used markers of autonomic nervous system dysfunction and can indicate an increased risk for the development of type 2 diabetes. However, in relation to CI and HRr in preschool children, several questions remain unanswered at present; therefore, this is an important issue for future research.

On the other hand, our results are in concordance with previous research on children, which has shown that the HR_peak_ is independent of sex, age, fitness level, and BMI [52,56,57]. Finally, contrary to one study [58], which observed a lower HR_Peak_ between overweight and obese children (aged 6–18 years), this work has been unable to find this relationship.

### 4.3. Cardiac Autonomic Function at Recovery after Exercise

In the current study, HRR was influenced by RHR, HR_peak_, HRr, CI, and HRV. It is noteworthy to indicate that sex does not influence HRR. Also, age showed a small impact on HRR. Likewise, Proudfoot et al. [54] did not show a relationship between HRR_1min_ and sex in preschool children, although they indicated higher values of HRR_1min_ (girls = 63 ± 15 bpm and boys = 67 ± 14 bpm) than those shown in the current study (girls = 54 ± 42 bpm and boys = 57 ± 57 bpm).

Regarding age influences, HRR was higher in older children. However, the findings of the current study do not support recent research that notes no relationship between HRR and age in preschool children [54]. Moreover, the present work found some significant correlations between RHR and HRV at rest with HRR_1min_ and HRR_5min_; these findings support the hypothesis that HRR immediately following exercise is related to resting parasympathetic modulation [59].

No relationship was found between HRR and BMI or WC. The findings are in the same line as those found by Easley et al. [37], which noted that obesity per se does not seem to have a significant impact on HRR. Additionally, other studies do not report any relationship between aerobic fitness and HRR_1min_ after a 20 m shuttle test in children aged 7 to 11 [60] and that HRR (1 min and 3 min) after submaximal exercise does not correlate with VO_2max_ in 10 years old children [46]. Corresponding with the previous reports, the current study did not find a relationship between endurance performance and HRR.

### 4.4. Heart Rate Variability

Although there are a few physiological factors that have an influence on HRV, such as age, sex, PA, weight status, and physical fitness [61], the current research did not find significant differences among sex in HRV. Nevertheless, these results differ from another published [62], which shows significantly greater HRV values in boys than in girls; these differences can be partially understood by differences in their RHR, which seems to be lower in boys [63], since modifications of the RHR may show a different impact on HRV [64]. Specifically, Crysarz et al. [65] indicated that 5- to 10-year-old boys show greater LF values than girls. However, some studies suggest contradictory findings in relation to the influence sex has on HRV [33].

Regarding age, a recent study reveals that the modifications of HRV in childhood consist of an increase in cardiac parasympathetic activity over sympathetic modulation [66]. In this study, older children exhibited higher HRV. These changes in HRV during childhood contemplate the heart’s capacity to react to physiological and environmental stimuli [21]. Furthermore, this work found results that showed that HRV has a strong association with HRr. These outcomes are in concordance with [67], which concludes that people who have a greater chronotropic competence and HRV values present a healthy cardiovascular status. [68]. In addition, this study did not find any links between HRV, WC, and BMI. This finding is supported by one study in which there is no relationship between weight status and CMR with HRV [69]. 

The main limitation of this study was its cross-sectional design. The present study did not consider designing a longitudinal investigation due to the difficulties of following preschoolers for years, given that most of them move to other schools when they grow up. Nevertheless, this study has an important strength since the sample comprises many children from a large region, including rural and nonrural areas. Moreover, it is also remarkable that this work provides important values related to autonomic cardiac function like RHR, HR_peak_, HRR, and HRV in infants. Therefore, to the best of our knowledge, this is the first study with these characteristics carried out on preschool children.

From a practical point of view and considering the lack of reference values for assessing the heart function during rest, maximal exercise, and recovery of Spanish preschool children, the values obtained in this study might play a key role for teachers, coaches, and physicians who work with 3–6-year-old children, which allows developing health programs in an individualized way, with these programs being based on measurable values of HR. In addition, assessing HR is inexpensive, and its easy use allows the test to be applied during both PA and clinical practice. In addition, the reference values can be used as a ‘warning signal’. Consequently, these values provide information about the necessity to conduct supplementary tests to determine the risk of developing cardiac diseases. This study proposes that further investigation is needed to assess the influence of PA participation on cardiorespiratory response in preschoolers.

## 5. Conclusions

This study concludes that the cardiovascular autonomic function at rest, exercise, and recovery in Spanish preschool children is not influenced by sex. In addition, this investigation also concludes that older children have greater cardiovascular modulation. Finally, this work enunciates that endurance capacity and weight status are not associated with HRV, RHR, HR_peak_, and HRR. In conclusion, the presented age and sex-specific reference range at rest, during exercise, and after exercise, of HR-related parameters determined by the 10 × 20 m test in Spanish children enable the assessment and monitoring of the cardiovascular system and the detection of children with low exercise tolerance or abnormal hemodynamic responses to exercise.

## Figures and Tables

**Figure 1 children-09-00654-f001:**
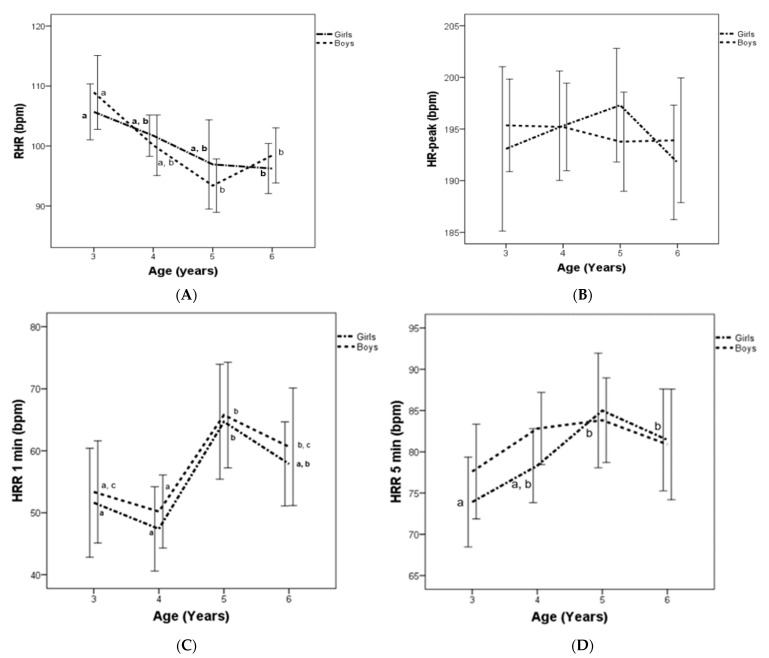
Resting heart rate (RHR) (**A**), HR_peak_ (**B**), and heart rate recovery (HRR) at 1 min (**C**) and 5 min (**D**) for Spanish boys and girls, by age group. Subscript letters indicate significant intra-group differences (*p* < 0.05).

**Figure 2 children-09-00654-f002:**
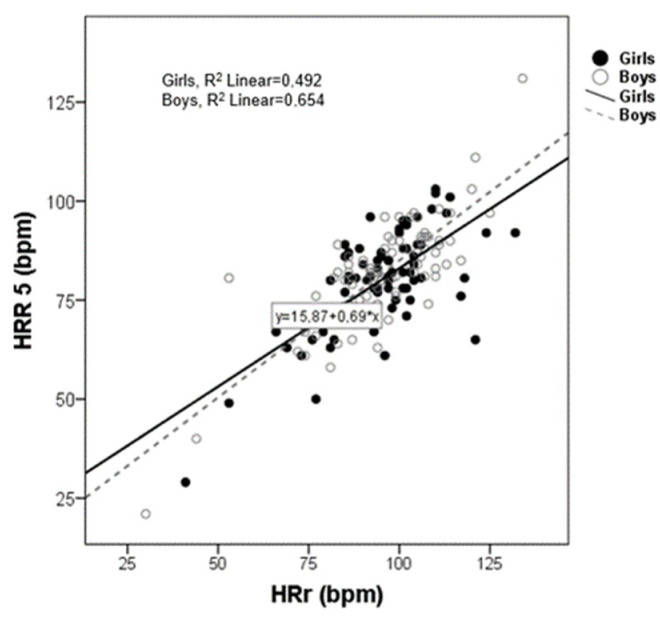
Scatter plot between heart rate reserve (HRr) and heart rate recovery (HRR) post exercise (5 min).

**Table 1 children-09-00654-t001:** Anthropometric characteristics, 10 × 20 m test, and heart rate response concerning sex and age groups.

	Girls Mean (SD)						Boys Mean (SD)					
Years	All n = 79	3 n = 16	4 n = 25	5 n = 13	6 n = 25	*p*-Value	All n = 88	3 n = 14	4 n = 25	5 n = 18	6 n = 31	*p*-Value
**Body mass (kg)**	20.51 (4.90)	15.61 (2.17)	19.88 (3.55)	21.69 (4.64)	23.69 (4.90)	<0.001	20.30 (4.55)	16.16 (4.84)	19.95 (2.72)	18.31 (3.27)	23.55 (4.21)	<0.001
**Height (cm)**	110.14 (14.60)	102.50 (9.21)	107.72 (5.20)	114.92 (6.71)	115.16 (22.36)	<0.001	109.28 (20.27)	100.86 (5.20)	110.56 (6.32)	114.65 (4.95)	109.10 (32.72)	<0.001
**BMI (kg/m^2^)**	16.40 (2.88)	14.97 (2.11)	17.18 (3.18)	16.50 (3.51)	16.49 (2.44)	0.108	15.81(2.76)	15.89 (4.16)	16.31 (1.72)	14.04 (2.91)	16.34 (2.26)	0.118
**WC (cm)**	56.94 (9.75)	54.27 (11.18)	60.84 (6.54)	55.0 (11.72)	55.16 (9.93)	0.047	52.29 (13.19) *	50.0 (12.87)	52.78 (13.81)	50.78 (13.24)	54.70 (13.19)	0.263
**RHR (bpm)≠**	100.00 (10.22)	105.69 (8.76)	101.72 (8.32)	96.92 (12.31)	96.24 (10.11)	0.014	99.55 (12.27)	108.93 (10.68)	100.12 (12.23)	93.39 (8.95)	98.42 (12.53)	0.003
**10 × 20 m (s)≠**	79.09 (18.35)	91.38 (11.21)	87.96 (14.26)	68.69 (16.00)	67.76 (17.58)	<0.001	74.84 (15.82)	96.07 (13.69)	77.00 (10.77) **	70.67 (11.30)	65.93 (13.14)	<0.001
**CI≠**	0.81 (0.11)	0.79 (0.12)	0.82 (0.11)	0.84 (0.07)	0.81 (0.12)	0.578	0.82 (0.11)	0.80 (0.08)	0.82 (0.07)	0.83 (0.08)	0.82 (0.15)	0.864
**HRr (bpm)≠**	94.06 (14.80)	87.37 (13.27)	93.60 (12.02)	100.3 (14.08)	95.52 (17.45)	0.114	94.93 (15.95)	86.43 (13.83)	95.08 (11.82)	100.39 (11.73)	95.48 (20.33)	0.104
**HR_Peak_ (bpm)**	194.07 (12.80)	193.0 (14.92)	185.32 (12.84)	197.31 (9.10)	191.77 (13.42)	0.663	194.48 (12.25)	195.36 (7.75)	195.20 (10.29)	193.78 (9.65)	193.91 (16.44)	0.914
**HRR_1min_(bpm)**	54.42 (17.15)	51.63 (16.49)	47.40 (16.51)	64.69 (15.32)	57.89 (16.46)	0.024	57.57 (20.25)	53.36 (14.27)	50.20 (14.28)	65.78 (17.12)	60.65 (25.85)	0.012
**HRR_5min_(bpm)**	79.51 (12.60)	73.91 (10.22)	78.32 (10.89)	85.00 (11.47)	81.45 (14.66)	0.026	81.53 (13.64)	77.61 (9.92)	82.82 (10.59)	83.83(10.31)	80.91 (18.30)	0.396
**RMSSD (ms)**	28.58 (10.61)	24.06 (9.54)	28.16 (9.5)	34.77 (13.26)	29.00 (6.03)	0.083	29.10 (13.15)	21.07 (8.28)	28.08 (10.84)	36.06 (15.16)	31.60 (16.62)	0.025
**SDNN (ms)**	54.28 (19.33)	44.69 (13.50)	53.24 (16.86)	68.69 (25.29)	53.00 (10.92)	0.036	56.82 (23.06)	41.93 (10.45)	57.40 (24.54)	66.00 (24.39)	62.60 (20.40)	0.012
**HF (ms^2^)**	1289.92 (935.1)	856.1 (566.4)	1256.6 (866.77)	1944 (1206)	1167.2 (718.07)	0.025	1270.10 (915.9)	852 (611)	1201 (972.48)	1614.92 (867.1)	1546.8 (1198.3)	0.079
**LF (ms^2^)**	1894 (1536.77)	1196.2 (942)	1770.5 (1844.9)	2496 (1630)	2151.2 (1334.8)	0.011	1896.5 (1416.8)	1023.34 (630.9)	1576.1 (1211.4)	2348.7 (1288.5)	2286.7 (1677.2)	0.003
**LF/HF**	176.83 (74.10)	167.6 (57.3)	166.8 (72.90)	154.6 (43.85)	204.30 (91.12)	0.295	199.42 (95.47)	184.8 (118)	189.44 (67.76)	190.47 (84.51)	218.94 (108.91)	0.334

SD: standard deviation. BMI: body mass index. WC: waist circumference. RHR: resting heart rate CI: chronotropic index HRr: heart rate reserve. HRR: heart rate recovery. RMSSD: square root of the mean squared differences of successive RR intervals. SDNN: standard deviation of all normal R-R intervals. HF: high-frequency band. LF: low-frequency band. LF/HF ratio: ratio of LF and HF frequency band powers. ≠ ANOVA. * Significant differences (*p* < 0.05) with girls. ** Significant differences (*p* < 0.01) with girls.

**Table 2 children-09-00654-t002:** Standardized beta coefficients from linear regression models for age, sex, anthropometric markers, performance in 10 × 20 m test, and HRV with cardiovascular autonomic function.

	RHR		HR_peak_		HRR_1min_		HRR_5min_	
	Beta	*p*-Valor	Beta	*p*-Valor	Beta	*p*-Valor	Beta	*p*-Valor
Age (Years)	−0.301	**<0.001**	−0.049	0.528	0.219	0.004	0.134	0.084
Sex	−0.020	0.796	0.016	0.833	0.084	0.283	0.077	0.326
BMI (Kg/m^2^)	−0.050	0.508	0.010	0.904	0.083	0.280	0.143	0.067
WC (cm)	−0.110	0.172	−0.051	0.551	−0.124	0.134	0.036	0.670
10 × 20 m (s)	0.081	0.382	−0.155	0.107	0.023	0.808	−0.032	0.523
RHR (bpm)			0.171	**0.036**	−0.251	**0.001**	−0.371	**<0.001**
HR_peak_ (bpm)					0.223	**0.003**	0.601	**<0.001**
HRr (bpm)	−0.561	**<0.001**	0.721	**<0.001**	0.361	**<0.001**	0.762	**<0.001**
CI	−0.021	0.782	0.966	**<0.001**	0.268	**<0.001**	0.672	**<0.001**
HRR_1min_(bpm)							0.272	**<0.001**
HRR_5min_(bpm)								
RMSSD (ms), rest	−0.799	**<0.001**	−0.191	**0.036**	0.233	**0.010**	0.334	**<0.001**
SDNN (ms), rest	−0.771	**<0.001**	−0.100	0.274	0.150	0.098	0.347	**<0.001**
HF (ms^2^), rest	−0.686	**<0.001**	−0.211	**0.020**	0.236	**0.009**	0.274	**0.002**
LF (ms^2^), rest	−0.604	**<0.001**	−0.077	0.324	0.260	**0.001**	0.263	**0.001**
LF/HF rest	0.243	**0.001**	−0.012	0.880	−0.055	0.483	−0.044	0.572

BMI: body mass index. WC: waist circumference. RHR: resting heart rate CI: chronotropic index. HRr: heart rate reserve. HRR: heart rate recovery. RMSSD: square root of the mean squared differences of successive RR intervals. SDNN: standard deviation of all normal R-R intervals. HF: high-frequency band. LF: low-frequency band. LF/HF ratio: ratio of LF and HF frequency band powers. Statistically significant *p*-values (*p* < 0.05) are in bold.

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
