# Peer review of "Cardiac Evaluation of Exercise Testing in a Contemporary Population of Preschool Children: A New Approach Providing Reference Values"

_children, 2022, doi:10.3390/children9050654_

Round 1

Reviewer 1 Report

I would like to thank the handling editor for giving me the opportunity to review this interesting study. I would also like to congratulate the authors for their thorough and informative paper.

The main purpose of this study was to develop a reference of the autonomic function during rest, maximal exercise, and in recovery using the 10 × 20 m test to screen CRF in healthy Spanish preschool children aged 3–6 years old, regarding age and sex

Here, I have made a few suggestions that, in my opinion, could help improve the overall quality of the manuscript.

W tym miejscu przedstawiłem kilka sugestii, które moim zdaniem mogą pomóc w poprawie ogólnej jakości rękopisu.

  • The authors may consider providing in full the abbreviations/acronyms in the Abstract, as they are not very commonly used, and readers may not be familiar with them.
  • Pozycje nr 2 i 3 w wykazie literatury uzupełnić o link do strony i czas dostępu

The authors may consider relating the obtained results to the developmental age of children,  not only to the chronological age.

How the children were motivated and encouraged to reach the best score possible in the test, please explain in the text.

Whether the RHR measurement was performed once or was there more than one measurement. Shouldn't the test be repeated?

Haven't you considered measuring the RHR while lying position?

Were the children allowed to do the test earlier?

Did the children know the test, wasn't it related to HR-Peak?

How many participants were excluded if the HR monitor got lost signal during the 10 × 20 m test.

Please describe at work how the running time was measured and whether the children were running individually or in a group?

Author Response

Point-by-point answer to reviewers’ comments

Manuscript ID children-1608274

Reviewer #1

We thank the reviewer for his/her encouraging comments. The authors really appreciate your time and your feedback. We believe we have properly addressed all your concerns and hope you will find this version suitable for publication. Below is a point-by-point answer to your comments. Changes to the original manuscript are highlighted in yellow. Please be aware that page numbers in this new version do not match those in the previous one.

Comments to the Author:

I would like to thank the handling editor for giving me the opportunity to review this interesting study. I would also like to congratulate the authors for their thorough and informative paper. The main purpose of this study was to develop a reference of the autonomic function during rest, maximal exercise, and in recovery using the 10 × 20 m test to screen CRF in healthy Spanish preschool children aged 3–6 years old, regarding age and sex.

Here, I have made a few suggestions that, in my opinion, could help improve the overall quality of the manuscript:

The authors may consider providing in full the abbreviations/acronyms in the Abstract, as they are not very commonly used, and readers may not be familiar with them.

Reply: Thank you for your comment. According to your suggestion, we have added an abbreviation section.

The authors may consider relating the obtained results to the developmental age of children, not only to the chronological age.

Reply: Thank you for your comment. According to the authors of this study, the concept of biological and chronological age is not important at preschool ages. However, in future studies we will consider this proposal made by the reviewer, given that the present study has not taken into consideration obtaining data on this subject.

How the children were motivated and encouraged to reach the best score possible in the test, please explain in the text.

Reply: Thank you for your comment. Basically, the children were motivated by two options. Firstly, the test was in the form of a game, so the child is always motivated. And secondly, the researchers who were conducting the test were constantly encouraging the children during each test.

Whether the RHR measurement was performed once or was there more than one measurement. Shouldn't the test be repeated?

Reply: Thank you for your comment. The tests were only carried out once. No test was performed twice.

Haven't you considered measuring the RHR while lying position?

Reply: Thank you for your comment. At first, we had in mind to measure the RHR in the lying position, but after reviewing the scientific literature we realized that most of the studies used this position because it might be more valid and reliable. Therefore, we decided to use the sitting position instead of the lying position (10.1139/h10-103).

Were the children allowed to do the test earlier?

Did the children know the test, wasn't it related to HR-Peak?

Reply: Thank you for your comment. The children did a walking test to familiarize themselves with the test. As for peak HR, the participants were too young to be taught concepts about cardiological factors.

How many participants were excluded if the HR monitor got lost signal during the 10 × 20 m test.

Reply: Thank you for your comment. Participants whose cardiac receptor lost signal during the test were excluded. The initial sample was 198 subjects; however, the final sample was 167 participants.

Please describe at work how the running time was measured and whether the children were running individually or in a group?

Reply: Thank you for your comment. The test was carried out on a group of 5 children. The researcher in charge of the experiment was responsible for taking the pupils to the school playground.  In addition, for each group of children there was a researcher in charge of taking the time of each child. The time was always recorded by the same researcher.

Reviewer 2 Report

Manuscript Number: CHI 1608274

Title: Comprehensive cardiac evaluation of exercise testing in a large contemporary population of preschool children: a new approach providing reference values

Main Strength and Weakness and Comments –

The manuscript is focused on to fulfill the lack of scientific evidence of the reference values of cardiac response (including maximal and rest) during the field exercise test in Spanish children aged 3-6y years old. The authors were especially interested in the linkage define the response over heart rate variability (cardiac autonomic function) in different periods from rest to exercise/recovery under 10 x 20m test.

The authors well constructed the structure of the manuscript. The autonomic response is an important aspect to assess in all periods of exercise. Even, clinically not being a cardiac concern, many children can have cardiac alteration without clinical presentation. This is can be an important aspect to consider in the prior assessment to exercise routine to children. However, regarding focus and scientific support in the introduction and discussion section, I have some worries. 

We feel that it has merit to be published at CHI and can be by one scientifically interesting manuscript, but it does not entirely meet CHI publishing criteria.

As the main weakness, I have to consider a too-long manuscript. In all sections, the authors could diminish the amount of word giving, more importantly, focus on this manuscript. It certainly will make this work readable to interested readers.

Besides this, in many parts of the writing, there is a lot of confusion regards to cardiorespiratory fitness (CRF) based on heart rate (rest, peak, and reserve) and variability of HR. Especially considering all those concepts are based on physiology and scientific pieces of evidence in adults. For instance, heart rate reserve is based on 220-age, for adults. In the same line, the measure of a load of exercise (TRIMP) is measured in the child of 3-6ys old. On the same hand, chronotropic index. My question is, even a good point to discuss, those aspects were not mentioned in the discussion. And how can we be sure these results are really expressive based on the strong discussion?

Another important aspect, the authors should think about is “Overall, there is strong evidence showing that RHR, HRpeak, HRR, and HRV may be considered an index of CRF and are useful tools for the assessment of cardiac autonomic function “. In my opinion, CRF is an important measure and marker of health cardiac in adults and teenagers, however, the narrow relation among those variables should take with care, especially those from cardiac autonomic function. I’m sure the one the most important aspect to assess health cardiac status, is the stability of cardiac autonomic function, however, in my opinion, CRF shouldn’t be considered as an index derived from HRV. Even physiologically related CRF is named by HR rest/peak/recovery, and not the stability of expression of R-R interval.

In several parts of the results and discussion section, is mentioned a maximal exercise test. What scientific parameter was used to define in your sample? And how did you define your data? I mean that this aspect makes those sections unpowered mainly considering the scope of CHI.

120L – Please clarify “…study was to develop a reference system of autonomic function at rest…” Did the authors develop a different system to assess cardiac autonomic function or only assessed cardiac autonomic function? It may be confused with a scientific paper.

L185-186 - How did you use intensity of exercise in TRIMP TEST? …which was calculated as proposed by Banister [45], this methodology incorporates the training duration, the mean HR of the total training session and the intensity of the exercise…”

My question is: is it reasonable to use the regression of TRIMP test measured from adults, with specific constant factors directed to adults with completely morphological, functional body and cardiovascular condition to children aged 3-6ys?

Finally, the authors certify at several moments of writing “reference values” however, it isn’t clear what the real values waited to those cardiac variables to 3-6 years old children.

Minor Comments:

Abstract

Please provide definition of HRV, HRr

Procedure:

L195 - Is there a real “experiment”?

L204 – Why did the authors register only 55% of HR? How did the authors work with the remaining 45% of HR data?

Methods

Results/table/figure

Table 1 needs edition. There are a lot of data occupying a different position of two variables at the same time. It is unreadable. Please move “(mean, SD)” from title to legend. The data inside the rows and columns were presented only (SD) in the parenthesis. 

What does mean letter a, b, c inside the images? And please provide on top of the figures the number, or letter to identify each one. In my point of view, it decreases the text in this section.  

Author Response

Point-by-point answer to reviewers’ comments

Manuscript ID children-1608274

Reviewer #2

We thank the reviewer for his/her encouraging comments. The authors really appreciate your time and your feedback. We believe we have properly addressed all your concerns and hope you will find this version suitable for publication. Below is a point-by-point answer to your comments. Changes to the original manuscript are highlighted in yellow. Please be aware that page numbers in this new version do not match those in the previous one.

Comments to the Author:

Title: Comprehensive cardiac evaluation of exercise testing in a large contemporary population of preschool children: a new approach providing reference values

Main Strength and Weakness and Comments –

The manuscript is focused on to fulfill the lack of scientific evidence of the reference values of cardiac response (including maximal and rest) during the field exercise test in Spanish children aged 3-6y years old. The authors were especially interested in the linkage define the response over heart rate variability (cardiac autonomic function) in different periods from rest to exercise/recovery under 10 x 20m test.

The authors well constructed the structure of the manuscript. The autonomic response is an important aspect to assess in all periods of exercise. Even, clinically not being a cardiac concern, many children can have cardiac alteration without clinical presentation. This is can be an important aspect to consider in the prior assessment to exercise routine to children. However, regarding focus and scientific support in the introduction and discussion section, I have some worries.

We feel that it has merit to be published at CHI and can be by one scientifically interesting manuscript, but it does not entirely meet CHI publishing criteria. As the main weakness, I have to consider a too-long manuscript. In all sections, the authors could diminish the amount of word giving, more importantly, focus on this manuscript. It certainly will make this work readable to interested readers.

Reply: Thank you for your comment. According to suggestion of review, we have reduced the number of words of the main document.

Besides this, in many parts of the writing, there is a lot of confusion regards to cardiorespiratory fitness (CRF) based on heart rate (rest, peak, and reserve) and variability of HR. Especially considering all those concepts are based on physiology and scientific pieces of evidence in adults. For instance, heart rate reserve is based on 220-age, for adults. In the same line, the measure of a load of exercise (TRIMP) is measured in the child of 3-6ys old. On the same hand, chronotropic index. My question is, even a good point to discuss, those aspects were not mentioned in the discussion. And how can we be sure these results are really expressive based on the strong discussion?

Reply: Thank you for your comment. We fully agree with the reviewer, trimp parameter is no specific to the pediatric population and it has been deleted for all analysis. Moreover, in the current study HRr was the difference between HR peak at exercise and HR at rest. In relation to resting heart rate, we provide the following information in the main document:

  • RHR may be useful for identifying children at increased risk of developing cardiovascular risk factors [57]. Specifically, children with a RHR equal to or greater than 91 beats per minute (bmp) present higher mean LDL cholesterol [14], and an increase of blood pressure, which are independent of adiposity, ethnicity, and age [15]. Otherwise, bradycardia, which is defined as a RHR that is slow compared to a normal RHR for a specific population’s age [16], is considered as a RHR of <60 bpm [17] and is related in pediatric patients with increased vagal tone, hypothyroidism, hypothermia, adrenal insufficiency, increased intracranial pressure, inherited arrhythmia, and cardiomyopathy or complete heart block and can lead to sudden death [18].
  • Although it may be difficult to define an optimal RHR for a given individual, according to Oschega et al., [54] and Hart [59], it seems desirable to maintain RHR among a recommended range of 86–109 bpm from ages 3 to 6. In the current study, although RHR was not recorded at basal condition, 67.6% of girls and 64.4% of boys were within these recommendation ranges.

  • In relation to HR peak and CI we also indicated that the clinical relevance of chronotropic incompetence in children is still unclear, and this is present many years before heart disease develops [22]. von Sheidt et al., [46] found values of CI < 0.80 in healthy children compared with children with congenital heart diseases, so it is advisable do not to use the threshold of 0.8 for the identification of chronotropic incompetence using treadmill exercise testing in children, most likely due to an overestimation of the maximal HR using the (220-age) formula.

  • Also, previous study with pediatric population have used this parameter (Von Scheidt, F., Meier, S., Krämer, J., Apitz, A., Siaplaouras, J., Bride, P., ... & Apitz, C. (2019). Heart rate response during treadmill exercise test in children and adolescents with congenital heart disease. Frontiers in pediatrics, 7, 65.)

  • We have added in the main paper that this is a controversial issue that should be analyzed in further studies.

Another important aspect, the authors should think about is “Overall, there is strong evidence showing that RHR, HRpeak, HRR, and HRV may be considered an index of CRF and are useful tools for the assessment of cardiac autonomic function “. In my opinion, CRF is an important measure and marker of health cardiac in adults and teenagers, however, the narrow relation among those variables should take with care, especially those from cardiac autonomic function. I’m sure the one the most important aspect to assess health cardiac status, is the stability of cardiac autonomic function, however, in my opinion, CRF shouldn’t be considered as an index derived from HRV. Even physiologically related CRF is named by HR rest/peak/recovery, and not the stability of expression of R-R interval.

Reply: Thank you for your comment. You are absolutely right; this sentence has been removed. Indeed, in the present study the 10x20m performance did not show significant correlations with any HR parameter.

In several parts of the results and discussion section, is mentioned a maximal exercise test. What scientific parameter was used to define in your sample? And how did you define your data? I mean that this aspect makes those sections unpowered mainly considering the scope of CHI.

Reply: Thank you for your comment.  According to Paridon et al., (10.1161/CIRCULATIONAHA.106.174375) a common criterion for determining whether the child performed a maximal effort was a maximum heart rate close to 200 bpm. This information has been added to the main document.

120L – Please clarify “…study was to develop a reference system of autonomic function at rest…” Did the authors develop a different system to assess cardiac autonomic function or only assessed cardiac autonomic function? It may be confused with a scientific paper.

Reply: Thank you for your comment. We only assessed cardiac autonomic function and added baseline values for RHR, HRpeak, HRR and HRV in this population.

L185-186 - How did you use intensity of exercise in TRIMP TEST? …which was calculated as proposed by Banister [45], this methodology incorporates the training duration, the mean HR of the total training session and the intensity of the exercise…”My question is: is it reasonable to use the regression of TRIMP test measured from adults, with specific constant factors directed to adults with completely morphological, functional body and cardiovascular condition to children aged 3-6ys?

Reply: Thank you for your comment. The reviewer is absolutely right. We very much appreciate the appreciation of this reviewer. These values were obtained directly from the VFC analysis software. Therefore, we have removed all TRIMP data from the main document.

Finally, the authors certify at several moments of writing “reference values” however, it isn’t clear what the real values waited to those cardiac variables to 3-6 years old children.

Reply: Thank you for your comment. We agree with you, maybe this expression is confusing; we wanted to indicate that the values obtained in the different heart rate parameters could be a reference for comparison in future studies.

Minor Comments:

Abstract

Please provide definition of HRV, HRr

Reply: Thank you for your comment. The definition of HRV and HRr have been included in abstract section.

Procedure:

L195 - Is there a real “experiment”?

Reply: Thank you for your comment. This is a mistake, the sentence “All the experiments were conducted in the school’s…” has been replaced by All tests were conducted in the school’s.….

L204 – Why did the authors register only 55% of HR? How did the authors work with the remaining 45% of HR data?

Reply: Thank you for your comment. First, we would like to apologies to the reviewer for the misunderstanding. This sentence was an error and comes from another previous study conducted by our research group. Please find attached the ANOVA table so that you can check that 100% of the HR data has been obtained for each participant.

N

Media

Desviación estándar

Error estándar

95% del intervalo de confianza para la media

Mínimo

Máximo

Límite inferior

Límite superior

MaxHR

0

79

194,07

12,880

1,449

191,19

196,96

149

215

1

88

194,48

12,254

1,306

191,88

197,08

140

215

Total

167

194,29

12,517

,969

192,37

196,20

140

215

HRreserva

0

79

94,0633

14,80241

1,66540

90,7477

97,3789

41,00

132,00

1

88

94,9318

15,94949

1,70022

91,5524

98,3112

30,00

134,00

Total

167

94,5210

15,37751

1,18995

92,1716

96,8703

30,00

134,00

MinHR

0

79

100,00

10,222

1,150

97,71

102,29

76

121

1

88

99,55

12,274

1,308

96,94

102,15

71

133

Total

167

99,76

11,318

,876

98,03

101,49

71

133

Descriptors

Methods

Results/table/figure

Table 1 needs edition. There are a lot of data occupying a different position of two variables at the same time. It is unreadable. Please move “(mean, SD)” from title to legend. The data inside the rows and columns were presented only (SD) in the parenthesis.

Reply: Thank you for your comment. According to your suggestion, we have edited the table 1.

What does mean letter a, b, c inside the images? And please provide on top of the figures the number, or letter to identify each one. In my point of view, it decreases the text in this section.

Reply: Thank you for your comment. Subscript letters indicate significant differences regarding age groups, in both boys and girls.
